# Is Evolutionary Conservation a Useful Predictor for Cancer Long Noncoding RNAs? Insights from the Cancer LncRNA Census 3

**DOI:** 10.3390/ncrna8060082

**Published:** 2022-12-07

**Authors:** Adrienne Vancura, Alejandro H. Gutierrez, Thorben Hennig, Carlos Pulido-Quetglas, Frank J. Slack, Rory Johnson, Simon Haefliger

**Affiliations:** 1Department of Medical Oncology, Inselspital, Bern University Hospital, University of Bern, 3010 Bern, Switzerland; 2Graduate School of Cellular and Biomedical Sciences, University of Bern, 3012 Bern, Switzerland; 3Department for BioMedical Research, University of Bern, 3008 Bern, Switzerland; 4HMS Initiative for RNA Medicine, Department of Pathology, Beth Israel Deaconess Medical Center Cancer Center, Harvard Medical School, Boston, MA 02215, USA; 5School of Biology and Environmental Science, University College Dublin, D04 V1W8 Dublin, Ireland; 6Conway Institute of Biomedical and Biomolecular Research, University College Dublin, D04 V1W8 Dublin, Ireland

**Keywords:** long noncoding RNA, cancer, conservation

## Abstract

Evolutionary conservation is a measure of gene functionality that is widely used to prioritise long noncoding RNAs (lncRNA) in cancer research. Intriguingly, while updating our Cancer LncRNA Census (CLC), we observed an inverse relationship between year of discovery and evolutionary conservation. This observation is specific to cancer over other diseases, implying a sampling bias in the selection of lncRNA candidates and casting doubt on the value of evolutionary metrics for the prioritisation of cancer-related lncRNAs.

## 1. Introduction

Long noncoding RNAs play central, functional roles in cancer and are being developed as targets for RNA therapeutics [1,2,3,4]. Given the high costs of drug discovery studies, and the frequency of late-stage failures, it is imperative to collect and effectively prioritise lncRNAs with the greatest therapeutic value. We here present the third version of the successful Cancer lncRNA Census (CLC3), covering publications from the period from 2019 to late 2020, comprising altogether 702 unique GENCODE-annotated lncRNAs with functional cancer roles based on a variety of evidence. 

## 2. Results

CLC3 incorporates and extends previous versions (CLC1, 118 lncRNAs, CLC2 374 lncRNAs) [1,2]. In addition to its size, CLC3 now incorporates for the first time lncRNAs involved in chemoresistance, with 10% of CLC lncRNAs exhibiting this functionality (Figure 1A). We previously observed that CLC lncRNAs carry a range of features distinguishing them from other lncRNAs. Amongst these were elevated rates for several measures of evolutionary conservation, similar to those previously observed for protein-coding cancer genes [1,5]. First, we evaluated the confidence level of experimental support for CLC genes, finding these to be consistent between versions with roughly 40% of lncRNAs validated by the highest-confidence in vivo evidence (Figure 1A). Therefore, any differences observed between CLC versions is not likely to arise from differences in confidence regarding their disease roles. Next, we comprehensively evaluated a range of features of CLC versions, comparing non-redundant gene sets to non-cancer lncRNAs (“nonCLC”) (Figure 1B). For comparison, we also compared a collection of disease-associated lncRNAs, from which cancer genes were removed (EVlncRNA) [6]. All CLC versions and EVlncRNAs display elevated levels of gene expression, expression ubiquity, overall gene length, spliced RNA length and proximity to nearest protein-coding genes (Figure 1B). Surprisingly, however, we noticed that CLC3 lncRNAs are not more evolutionarily conserved compared to other non-cancer lncRNAs (arrows). This is true not only for two different measures of conservation from the widely used PhastCons measure (average base-level score and percentage of exon coverage by conserved elements), but also for the promoter (average base-level), for which particularly elevated conservation has been observed in lncRNAs [7,8]. A more detailed gene-level inspection supported these findings (Figure 1C–F), showing a pronounced trend for the CLC3 lncRNAs to have comparable or even lower conservation than lncRNAs in general.

To strengthen these findings, we used an alternative method to evaluate evolutionary conservation: the existence of orthologous lncRNA genes in other species. Using the tool ConnectOR [9], we searched for orthologues of human lncRNAs in chimpanzees and mice (see Methods). Overall, we identified orthologues for 4102 and 4493 lncRNAs in chimpanzees and mice, respectively (lower rates in chimpanzees likely reflect less mature lncRNA annotations). Consistent with previous results, we observed that CLC3 lncRNAs have a significantly lower chance of having an identifiable orthologue than CLC1 and CLC2, at a level comparable to nonCLC lncRNAs (Figure 2A). 

Given that CLC3 lncRNAs were collected most recently, we hypothesised that the observed trend arose from a relationship between conservation and the moment when the lncRNA was studied. Indeed, we observed a significant negative correlation between conservation and year of discovery (Figure 2B, left). This trend appears to be specific to cancer, because EVlncRNAs from other diseases do not display this behaviour (Figure 2B, right). In other words, as time goes on, researchers are turning their attention to less conserved lncRNAs that nevertheless play functional cancer roles.

## 3. Discussion

In summary, we have presented the latest version of the Cancer lncRNA Census, a carefully curated resource of functional cancer-associated lncRNAs intended to serve as a useful true positive dataset for large-scale discovery and as a source for therapeutic development. We have made the surprising observation that evolutionary conservation of collected lncRNAs decreases with year of publication, and that recently published cancer lncRNAs have conservation levels similar to lncRNAs in general. Although previous studies have shown that protein-coding cancer genes are conserved more on average [1], it remains possible that a similar phenomenon affects these genes. This phenomenon appears to be specific for cancer, since catalogues of lncRNAs playing roles in other diseases do not display the same trend. Evolutionary conservation is a longstanding and widely used criterion for the selection of candidate lncRNAs for follow-up study [10,11,12,13]. However, there are numerous examples of functionally validated, non-conserved lncRNAs [14,15]. Supporting these findings, recent unbiased large-scale functional screens found no relationship between conservation and hits [16]. The apparent specificity to cancer raises the possibility that tumours exploit lncRNA sequences that have no natural function. Indeed, a similar model was recently proposed by Adnane and colleagues [17]. These findings suggest that the scientific community may have suffered an unconscious bias in selecting evolutionarily conserved lncRNAs for study, thereby reinforcing the impression that conservation is a useful criterion for candidate selection [1]. This work suggests that filtering by evolutionary conservation may result in omission of important cancer related lncRNAs.

## 4. Materials and Methods

### 4.1. Literature Search and LnCompare for Feature and Repeat Analysis

This analysis was performed as described in CLC2 [2] and the full CLC3 gene list can be found here: https://zenodo.org/record/7075104#.YyCB0C1Q3T8 (accessed on 13 September 2022).

### 4.2. EVlncRNA Non-Cancer lncRNA Dataset

EVlncRNAs were downloaded from and were sorted for ENSG (GENCODE v28) and overlayed with CLC genes to exclude functional cancer lncRNAs.

### 4.3. Conservation Scores

Exons were collapsed using exon info from GENCODE v28, and PhastCons exon conservation scores (PhastCons100way.UCSC.hg28) were generated according to Vancura et al., 2021 using Bioconductor Genomic Scores R package. 

PhastConsElements 100way were downloaded from genome.ucsc.edu using the table browser. PhastConsElements were intersected with datasets using intersectBed. Statistical evaluation was performed using Wilcoxon test.

### 4.4. Publication Year

PMID years for each lncRNA were extracted using the code from https://www.ncbi.nlm.nih.gov/books/NBK179288/ (accessed on 5 March 2022) and earliest publication years were used for subsequent analysis. 

### 4.5. Orthologue Prediction 

Orthologue prediction was performed using ConnectOR (https://github.com/Carlospq/ConnectOR (accessed on 20 May 2022)) based on LiftOver of syntenic regions from human (hg38) to mouse (mm10) or chimpanzee (panTro3). ConnectOR results “not lifted” and “one to none” were characterised as no orthology prediction. Statistical evaluation was performed using Fisher’s one-sided *t*-test.

## Figures and Tables

**Figure 1 ncrna-08-00082-f001:**
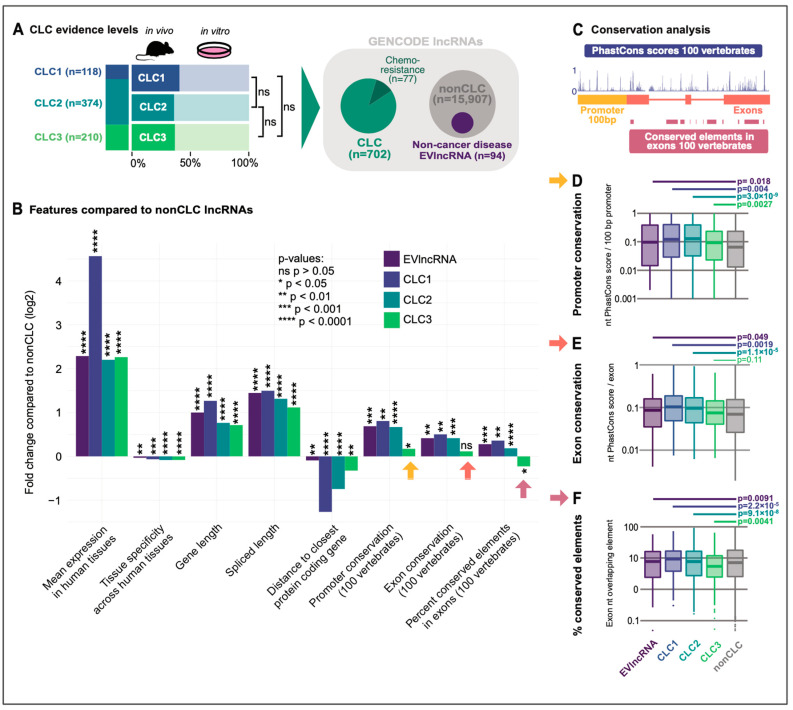
(**A**) Evidence levels for functional lncRNAs in CLC database versions. Dark colour indicates number of lncRNAs tested in an in vivo setting. No significant (ns) difference of in vivo enrichment is observed across the datasets. The full CLC consists of 702 lncRNAs with 77 lncRNAs exhibiting chemoresistance mechanisms. GENCODE lncRNAs are subdivided in CLC and nonCLC genes for further comparison. Non-cancer disease EVlncRNAs are nonCLC genes indicating a disease functionality but not represented in the CLC database. (**B**) Features in datasets compared to nonCLC lncRNAs using LnCompare. (**C**) Overview of conservation analysis using 100 vertebrates comparisons. (**D**) Promoter conservation analysis for all datasets. (**E**) Exon analysis for all datasets. (**F**) Conserved elements analysis for all datasets.

**Figure 2 ncrna-08-00082-f002:**
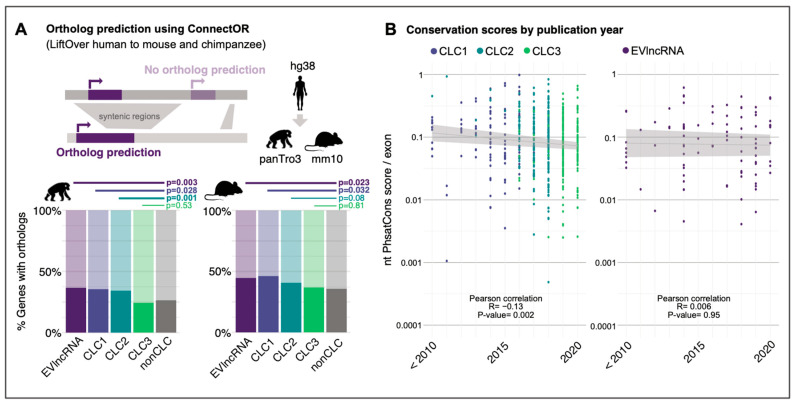
(**A**) Ortholog prediction using ConnectOR for chimpanzees (left) and mice (right). (**B**) Exon conservation scores by publication year for CLC versions (left) and EVlncRNAs (right).

## Data Availability

The CLC3 gene list can be found here: https://zenodo.org/record/7075104#.YyCB0C1Q3T8.

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
