# Peer review of "Is Evolutionary Conservation a Useful Predictor for Cancer Long Noncoding RNAs? Insights from the Cancer LncRNA Census 3"

_ncrna, 2022, doi:10.3390/ncrna8060082_

Round 1

Reviewer 1 Report

This study by Vancura et al. presented the updated, 3rd version of Cancer lncRNA Census (CLC3), and analyzed the difference between the lncRNAs in CLC3 and those in the previous versions (CLC1-2) in terms of evidence levels and promoter/exon conservations. The authors noticed that lncRNAs in CLC3 have lower conservation levels compared to lncRNAs in general, allowing them to assume that not-conserved lncRNAs should be also considered as candidates of potential functional molecules in cancer pathobiology. The data suggests the assumption, and I support publication of this manuscript. However, I question whether they can clearly say “evolutionary conservation is not a useful filter when selecting cancer lncRNAs for further study,” because there is no comparison of “how important the lncRNAs are (extent/degree/magnitude of their importance)” between conserved and not-conserved lncRNAs. What if conserved lncRNAs have more significant functions in cancer than not-conserved lncRNAs? Because this study only considered functional/non-functional point of view, the authors might want to soften their claims.    

Author Response

Thank you for the insightful suggestion. Indeed, we do not compare conservation to significant function of cancer lncRNAs as to our notice, there is no available scaling system at the moment. We do however agree to soften the claim and adjusted the manuscript accordingly: “Our observation leads to the assumption that lncRNAs with lower evolutionary conservation are functional in cancer cells and should be considered for in-depth studies.”

Reviewer 2 Report

The paper Is evolutionary conservation a useful predictor for cancer long noncoding RNAs? Insights from the Cancer lncRNA Census 3 present an important consideration to select lncRNAs involved in cancer. I recommend to be published before minor correction, please revised the citation style.  

Author Response

Thank you for your recommendation. The citation style was revised.

Reviewer 3 Report

In the present study, the authors used their previously established lncRNA tool kit to logically reason a new, although negative finding: evolutionary conservation may not be a useful filter in the selection of cancer lncRNAs. In my opinion, as a potential reader of the journal, the present study is relatively solid and meaningful, and the story is quite intact, but a more comprehensive introduction of the tool kit that the authors developed in the published papers should be made. 

One more concern was raised as I noticed the type of the manuscript is brief report shown in the system but an original article shown on the pdf script. I believe the current form of the manuscript is ready to publish as brief report, indeed, but an extensive re-writing would be recommended if it is going to be an original article (and once again, in this case, introduce more about your previous work on this tool kit). 

In a nutshell, principally speaking, the author did a great job which worth publishing. 

Author Response

Thank you very much for your suggestions. This manuscript was intended to be published as a brief report and the pdf script was adjusted accordingly.